# Age-Stratified Analysis of First-Line Chemoimmunotherapy for Extensive-Stage Small Cell Lung Cancer: Real-World Evidence from a Multicenter Retrospective Study

**DOI:** 10.3390/cancers15051543

**Published:** 2023-02-28

**Authors:** Takayuki Takeda, Tadaaki Yamada, Yusuke Kunimatsu, Keiko Tanimura, Kenji Morimoto, Shinsuke Shiotsu, Yusuke Chihara, Asuka Okada, Shigeto Horiuchi, Makoto Hibino, Kiyoaki Uryu, Ryoichi Honda, Yuta Yamanaka, Hiroshige Yoshioka, Takayasu Kurata, Koichi Takayama

**Affiliations:** 1Department of Respiratory Medicine, Japanese Red Cross Kyoto Daini Hospital, Kyoto 602-8026, Japan; 2Department of Pulmonary Medicine, Graduate School of Medical Science, Kyoto Prefectural University of Medicine, Kyoto 602-8566, Japan; 3Department of Respiratory Medicine, Japanese Red Cross Kyoto Daiichi Hospital, Kyoto 605-0981, Japan; 4Department of Respiratory Medicine, Uji-Tokushukai Medical Center, Kyoto 611-0041, Japan; 5Department of Respiratory Medicine, Saiseikai Suita Hospital, Osaka 564-0013, Japan; 6Department of Respiratory Medicine, Shonan Fujisawa Tokushukai Hospital, Kanagawa 251-0041, Japan; 7Department of Respiratory Medicine, Yao Tokushukai General Hospital, Osaka 581-0011, Japan; 8Department of Respiratory Medicine, Asahi General Hospital, Chiba 289-2511, Japan; 9Department of Thoracic Oncology, Kansai Medical University Hospital, Osaka 573-1191, Japan

**Keywords:** age-stratified analysis, chemoimmunotherapy, elderly patient, post-progression survival (PPS), prognostic nutritional index (PNI), small cell lung cancer

## Abstract

**Simple Summary:**

Chemoimmunotherapy improved overall survival (OS) and progression-free survival (PFS) in patients with extensive-stage small cell lung cancer (ES-SCLC) in two phase III trials, which set the age-stratified subgroup analyses at 65 years. Considering the super-aged society of Japan, treatment efficacy and safety in elderly patients ≥ 75 years with ES-SCLC should be validated through real-world Japanese evidence. Consecutive 225 Japanese patients with SCLC were evaluated, and 155 received chemoimmunotherapy (98 non-elderly and 57 elderly patients). The dose reduction at initiating the first cycle was significantly higher in the elderly (47.4%) than in the non-elderly (20.4%) patients (*p* = 0.03). The median PFS and OS in the non-elderly and the elderly were 5.1 and 14.1 months and 5.5 and 12.0 months, respectively, without significant differences. Multivariate analyses revealed that age, the baseline Eastern Cooperative Oncology Group performance status, and dose reduction at initiating the first chemoimmunotherapy cycle were not correlated with PFS or OS.

**Abstract:**

Chemoimmunotherapy improved overall survival (OS) and progression-free survival (PFS) in patients with extensive-stage small cell lung cancer (ES-SCLC) in two phase III trials. They set the age-stratified subgroup analyses at 65 years; however, over half of the patients with lung cancer were newly diagnosed at ≥75 years in Japan. Therefore, treatment efficacy and safety in elderly patients ≥ 75 years with ES-SCLC should be evaluated through real-world Japanese evidence. Consecutive Japanese patients with untreated ES-SCLC or limited-stage SCLC unfit for chemoradiotherapy between 5 August 2019 and 28 February 2022 were evaluated. Patients treated with chemoimmunotherapy were divided into the non-elderly (<75 years) and elderly (≥75 years) groups, and efficacy, including PFS, OS, and post-progression survival (PPS) were evaluated. In total, 225 patients were treated with first-line therapy, and 155 received chemoimmunotherapy (98 non-elderly and 57 elderly patients). The median PFS and OS in non-elderly and elderly were 5.1 and 14.1 months and 5.5 and 12.0 months, respectively, without significant differences. Multivariate analyses revealed that age and dose reduction at the initiation of the first chemoimmunotherapy cycle were not correlated with PFS or OS. In addition, patients with an Eastern Cooperative Oncology Group performance status (ECOG-PS) = 0 who underwent second-line therapy had significantly longer PPS than those with ECOG-PS = 1 at second-line therapy initiation (*p* < 0.001). First-line chemoimmunotherapy had similar efficacy in elderly and non-elderly patients. Individual ECOG-PS maintenance during first-line chemoimmunotherapy is crucial for improving the PPS of patients proceeding to second-line therapy.

## 1. Introduction

Lung cancer is the leading cause of mortality in cancer patients, of which, small cell lung cancer (SCLC) accounts for approximately 10–15% [1]. Due to SCLC’s aggressive and devastating nature, approximately two-thirds of patients present with extensive-stage SCLC (ES-SCLC), whose treatment goal is to prolong overall survival (OS) without expecting a complete cure [2]. The standard treatment has been platinum doublet chemotherapy with an OS of 9.3–12.8 months [3,4,5]. However, introducing immune checkpoint inhibitors (ICIs), especially anti-programmed cell death ligand 1 (PD-L1) monoclonal antibodies, has changed ES-SCLC outcomes. Chemoimmunotherapy consisting of atezolizumab combined with carboplatin (CBDCA) and etoposide (ETP) significantly improved OS and progression-free survival (PFS) compared to the placebo plus CBDCA and the ETP group in the IMpower133 phase III trial reported in 2018 [6]. Another chemoimmunotherapy consisting of durvalumab plus cisplatin (CDDP) or CBDCA and ETP significantly improved OS and PFS in the CASPIAN phase III trial [7].

The pre-specified age-stratified subgroup analyses in these phase III trials were set at 65 years, and the observed adverse events (AEs) and immune-related AEs (irAEs) were tolerable among patients aged ≥ 65 years [6,7]. However, an exponential increase in the number of elderly patients with lung cancer has been noticed worldwide in recent years owing to population aging in developed countries, which is the cardinal risk of malignancy. In the United States, the median age of those patients diagnosed with lung cancer is 70 years [8], and over 35% of patients diagnosed with lung cancer are aged > 75 years in the U.S. based on the Surveillance, Epidemiology, and End Results (SEER) database [9]. In Japan, the super-aged society is becoming the oldest worldwide, with 28.4% aged ≥ 65 and 14.7% ≥ 75 [10]. Over half of the newly diagnosed lung cancer patients were >75 years [11]. Thus, the efficacy and safety of chemoimmunotherapy for elderly patients with ES-SCLC aged ≥ 75 should be validated through real-world Japanese evidence because there are no prospective phase III trials in this population. Only one prospective observational study with a small sample size has assessed this population [12].

The limited efficacy of ICIs and the increased risk of irAEs among elderly patients with non-small cell lung cancer (NSCLC) have been reported [13]. The accumulated evidence of the restricted efficacy of ICIs among elderly patients with NSCLC [14] could be partly explained by immunosenescence [13,15]. In contrast, there is scarce evidence regarding the efficacy and safety of ICIs in elderly patients with ES-SCLC. Therefore, exploring whether elderly patients aged ≥ 75 with ES-SCLC would benefit from the standard first-line treatment of chemoimmunotherapy equivalent to their younger counterparts is useful.

Furthermore, biomarkers for predicting OS and PFS in patients with ES-SCLC during chemoimmunotherapy remain scarce. The pretreatment platelet-to-lymphocyte ratio (PLR) in patients with ES-SCLC receiving first-line chemotherapy reportedly correlated with OS and PFS in a phase II trial cohort [16]. Similarly, some immunological and nutritional markers during platinum doublet chemotherapy have correlated with ES-SCLC prognosis, including the neutrophil-to-lymphocyte ratio (NLR) in a meta-analysis [17] and a single institute [18], the prognostic nutritional index (PNI) in a meta-analysis [19], PNI in combination with neuron-specific enolase [20], and the systemic immune-inflammation index (SII) in a single institute [21]. However, a contradictory report from a single institute exists where NLR did not correlate with OS in patients with SCLC [22].

Thus, this multicenter retrospective study aimed to investigate the efficacy and safety of first-line chemoimmunotherapy for patients ≥ 75 years with ES-SCLC compared to their younger counterparts and identify baseline characteristics and immunological and nutritional markers that could predict OS and PFS. This study also evaluated post-progression survival (PPS) after first-line chemoimmunotherapy, which was highly correlated with OS compared to PFS in patients with ES-SCLC who were treated with chemoimmunotherapy [23]. PPS evaluation was added to elucidate the efficacy of second-line or later therapy in patients with ES-SCLC. To our knowledge, this is the first study on the age-stratified analysis of chemoimmunotherapy in patients with ES-SCLC aged ≥ 75.

## 2. Materials and Methods

### 2.1. Patients

This multicenter retrospective study evaluated consecutive Japanese patients with untreated ES-SCLC or limited-stage SCLC (LS-SCLC) who were unfit for chemoradiotherapy at nine hospitals in Japan between 5 August 2019 (the approval date of the first-line CBDCA plus ETP combined with atezolizumab in Japan) and 28 February 2022. The data cut-off date was 31 August 2022, with a minimum follow-up of 6 months after the final enrolment date.

The inclusion criteria were: (1) cytologically or histologically diagnosed ES-SCLC or LS-SCLC unfit for curative radiotherapy and (2) patients treated with first-line chemoimmunotherapy (CBDCA + ETP + atezolizumab, CBDCA+ ETP + durvalumab, CDDP + ETP + durvalumab) or platinum doublet chemotherapy (CBDCA + ETP, CDDP + ETP, and CDDP + irinotecan), without treatment history.

This study aimed to elucidate the efficacy and safety of chemoimmunotherapy for elderly patients with ES-SCLC. The chemoimmunotherapy group was divided into non-elderly (<75 years) and elderly (≥75) groups. The data of patients treated with platinum doublet chemotherapy were not compared with the chemoimmunotherapy counterparts because those who did not receive chemoimmunotherapy may have some characteristics unfit for ICIs, leading to bias and strong confounding. Therefore, the reasons they did not receive chemoimmunotherapy were recorded in the platinum doublet group, and age-stratified analyses were conducted among the chemoimmunotherapy group.

The study protocol was in accordance with the Declaration of Helsinki and was approved by the Ethics Committees of the Japanese Red Cross Kyoto Daini Hospital (5 August 2022; S2022-16) and of each participating hospital. The requirement for informed consent was waived based on the retrospective analysis of anonymized patient data. Patients were allowed to opt out of their data use, and the relevant information was publicly available on each hospital’s website.

### 2.2. Treatment with First-Line Chemoimmunotherapy and Response Evaluation

All patients were treated with 1–6 cycles of induction chemoimmunotherapy, followed by maintenance therapy with the relevant anti-PD-L1 antibody until disease progression, death, or unacceptable toxicity. The administered regimens were: (1) ETP (80–100 mg/m^2^ body surface area, administered intravenously on days 1–3 of each cycle), CBDCA (area under the curve of 5–6 mg/mL per min, administered intravenously on day 1 of each cycle), and atezolizumab (fixed dose of 1200 per body, administered intravenously on day 1 of each cycle), followed by maintenance atezolizumab 1200 mg every three weeks; (2) platinum-ETP consisting of ETP (80–100 mg/m^2^ on days 1–3 of each cycle) combined with the physicians’ choice of CBDCA (area under the curve of 5–6 mg/mL per min) or CDDP (75–80 mg/m^2^) intravenously administered on day 1 of each cycle, and durvalumab (fixed dose of 1500 mg per body, administered intravenously on day 1 of each cycle), followed by maintenance durvalumab 1500 mg every 4 weeks. The standard dosage of ETP, CBDCA, and CDDP had been previously defined at each participating hospital, and dose reduction was each treating physician’s choice.

The response was evaluated according to the best overall treatment response using the response evaluation criteria in solid tumours version 1.1 [24]. The treating physicians and radiologists used chest radiography, computed tomography, and brain magnetic resonance imaging with contrast enhancement to evaluate the treatment response. The responses were classified as complete response (CR), partial response (PR), stable disease (SD), progressive disease (PD), and not evaluable (NE).

The PPS was the period from PD for first-line chemoimmunotherapy to any-cause death or the last follow-up in censored cases without death. The correlations between PFS, OS, PPS, and OS were evaluated.

The patient’s Eastern Cooperative Oncology Group performance status (ECOG-PS) was documented at baseline, at the time of disease progression, and at second-line therapy initiation if available. The relationship between ECOG-PS at the time of disease progression and PPS after chemoimmunotherapy was investigated. Thereafter, the correlation between ECOG-PS at second-line therapy initiation and PPS after chemoimmunotherapy was also investigated to indirectly elucidate the importance of maintaining ECOG-PS during the first-line chemoimmunotherapy.

Treatment beyond disease progression was allowed when there was a clinical benefit, where the PPS was estimated from the end of beyond disease progression. Prophylactic cranial irradiation was allowed at the physician’s discretion.

### 2.3. Data Collection

Baseline characteristics, namely, age, sex, smoking history, ECOG-PS score, comorbidities, height, body weight, histology, blood investigations, systemic corticosteroid administration at baseline, regimen administered, dosage (standard or reduced) at initiating the first induction chemoimmunotherapy cycle, dose reduction during subsequent cycles, usage of granulocyte colony-stimulating factor (G-CSF), efficacy including objective response rate (ORR), disease control ratio (DCR), PFS, OS, and PPS, were reasons for discontinuation of first-line chemoimmunotherapy, observed AEs including irAEs. Subsequent therapy options were collected from electronic medical records. AEs were evaluated using the common terminology criteria for adverse events (version 5.0).

### 2.4. Study Variables

Previous reports were used to define cut-off values for the following immunological and nutritional markers: PLR [16,25,26], NLR [17,18,21,22,26,27], PNI [19,20,26,28], and SII [21,29,30]. PLR was the ratio of absolute platelet count (/µL) divided by absolute lymphocyte count (/µL) and grouped based on PLR < 250 or ≥250. NLR was the ratio of absolute neutrophil count (/µL) to lymphocyte count (/µL) and grouped using NLR < 5 or ≥5. The PNI was calculated as 10 × albumin (g/dL) + 0.005 × absolute lymphocyte count (/µL), and grouped using PNI ≥ 40 or <40. The SII values were calculated as absolute platelet count (/L) × NLR and grouped according to SII values of <1000 or ≥1000.

These markers were evaluated at baseline for associations with PFS and OS.

### 2.5. Statistical Analysis

Statistical analyses were performed using EZR [31], a graphical user interface for R software (The R Foundation for Statistical Computing, Vienna, Austria). Baseline characteristics were compared using Pearson’s Chi-square test. Median PFS and OS intervals with corresponding 95% confidence intervals (CIs), ORR, and DCR were calculated. Curves for PFS and OS were illustrated using the Kaplan–Meier method and log-rank tests. Univariate and multivariate analyses were performed for each potential marker or baseline characteristic using Cox proportional hazards regression analyses, in which hazard ratios (HR) and 95% CIs were estimated. Spearman’s rank correlation and linear regression analyses were used to evaluate the correlations between PFS or PPS and OS via a correlation coefficient (r_s_ value), *p*-value, and linear regression (R^2^ value). Differences were considered significant at *p* < 0.05.

## 3. Results

### 3.1. Patient Selection Diagram (Figure 1) and the Characteristics of Patients Treated with Platinum Doublet Chemotherapy (Table 1)

In total, 225 patients with ES-SCLC were treated with first-line chemoimmunotherapy or platinum doublet chemotherapy, including 178 males and 47 females with a median age of 73 (range: 43–87). Among the 225 patients, 70 were treated with platinum doublet chemotherapy, and 155 received first-line chemoimmunotherapy.

The 70 patients in the chemotherapy group included 53 males and 17 females, and 26 non-elderly and 44 elderly patients with a median age of 76 (range: 61–86); 6 (23.1%) out of 26 non-elderly patients received CDDP-based regimens which were not administered in elderly patients (*p* = 0.04). On the other hand, all elderly patients received CBDCA plus ETP, which was selected in 20 out of 26 non-elderly patients (*p* = 0.002). There was no significant difference in the baseline characteristics between the non-elderly and elderly patients. The dominant reasons why chemoimmunotherapy was considered unfit and chemotherapy was adopted were underlying interstitial lung disease, poor ECOG-PS, age, patient preference, and underlying autoimmune disease (Table 1).

### 3.2. Characteristics of Patients Treated with Chemoimmunotherapy (Table 2)

The 155 patients treated with chemoimmunotherapy included 98 non-elderly and 57 elderly patients, with a median age of 72 (Table 2). The dose reduction at initiating the first cycle was significantly higher in the elderly (47.4%) than in the non-elderly (20.4%) patients (*p* = 0.03). Among the 155 patients, 13 (8.4%) were still on the first-line chemoimmunotherapy at the data cut-off, 99 (63.9%) underwent subsequent second-line treatment, and 43 (27.7%) selected the best supportive care without receiving second-line treatment.

### 3.3. Objective Response Rate, Progression-Free Survival, and Overall Survival Outcomes (Figure 2 and Figure 3)

The median follow-up for the entire study population was 11.5 months (range: 6.0–37.3). The treatment responses to chemoimmunotherapy in the 155 patients were classified as CR in 2 (1.3%), PR in 124 (80.0%), SD in 20 (12.9%), PD in 5 (3.2%), and NE in 4 (2.6%) patients, resulting in an ORR of 81.3% and DCR of 94.2% (Appendix A). The median PFS and OS were 5.2 months (95% CI: 4.8–5.5) and 13.2 months (95% CI: 11.7–15.5), respectively (Figure 2).

Next, we performed age-stratified analyses. The treatment responses in 98 non-elderly patients were CR in 2 patients (2.0%), PR in 77 (78.6%), SD in 14 (14.3%), PD in 4 (4.1%), and NE in 1 (1.0%), resulting in an ORR of 80.6% and DCR of 94.9%. In contrast, the 57 elderly patients had CR in 0, PR in 47 (82.5%), SD in 6 (10.5%), PD in 1 (1.8%), and NE in 3 (5.3%) patients, resulting in an ORR of 82.5% and DCR of 93.0%. The median PFS in non-elderly and elderly patients was 5.1 months (95% CI: 4.7–5.5) and 5.5 months (95% CI: 4.6–6.4), respectively. The median OS in non-elderly and elderly patients was 14.1 months (95% CI: 11.7–17.0) and 12.0 months (95% CI: 8.3–16.9), respectively. There were no significant differences between both groups regarding ORR (*p* = 0.83), DCR (*p* = 0.73), PFS (*p* = 0.19), and OS (*p* = 0.59).

The Kaplan–Meier curves for PFS and OS in each group are illustrated in Figure 3A,B, respectively.

### 3.4. Relationship between Overall Survival and Progression-Free Survival or Post-Progression Survival after First-Line Chemoimmunotherapy (Figure 4)

The correlation between PFS and OS is presented in Figure 4A, and that between PPS and OS is indicated in Figure 4B. Spearman’s rank correlation analysis revealed a moderate correlation between PFS and OS (r_s_ = 0.60, *p* < 0.001, R^2^ = 0.48), whereas PPS and OS displayed a significantly higher correlation (r_s_ = 0.86, *p* < 0.001, R^2^ = 0.73).

### 3.5. Adverse Events and Dose Reduction during Induction Therapy

AEs of any grade were observed in 143 patients (92.3%), including 90 non-elderly (91.8%) and 53 elderly (93.0%) patients (Appendix A). In total, 12 patients (7.7%) of the total study population experienced adverse events leading to treatment withdrawal, including 6 non-elderly (6.1%) and 6 elderly (10.5%) patients. The most common AE was neutropenia, occurring in 114 (73.5%) of the total population, in 68 (69.4%) non-elderly and 46 (80.7%) elderly patients.

IrAEs of any grade were observed in 30 patients (19.4%). Adrenal insufficiency (grade 1) and colitis (grade 1) were observed in the same elderly patient. All irAEs of ≥grade 3, excluding hyperthyroidism, and one patient who experienced grade 2 pneumonitis led to treatment withdrawal: one with grade 2, two with grade 3, and one with grade 4 pneumonitis, one with grade 4 myasthenia gravis, one with grade 3 encephalopathy, and one with grade 3 vasculitis. No grade 5 AE was observed.

There was no difference between the groups regarding AEs of all grades (*p* = 0.78), AEs ≥ grade 3 (*p* = 0.11), and those leading to discontinuation (*p* = 0.36), and between both groups in dose reduction during induction (*p* = 0.176).

### 3.6. Relationship between the Baseline Characteristics or Candidate Predictive Biomarkers and Progression-Free Survival or Overall Survival among 155 Patients (Table 3), Elderly Patients Aged ≥ 75 (Table 4), and Non-Elderly Patients < 75 Years (Table 5)

Univariate analyses were performed to predict PFS and OS using baseline characteristics and predictive markers (Table 3). PFS outcomes were significantly associated with a baseline PNI of ≥40 vs. <40 (HR: 0.64 [95% CI: 0.45–0.92], *p* = 0.02) and the number of induction chemoimmunotherapy cycles (4–6 vs. 1–3 cycles; HR: 0.38 [95% CI: 0.25–0.95], *p* < 0.001). OS outcomes were significantly associated with ECOG-PS of 0–1 vs. ≥2 (HR: 0.43 [95% CI: 0.25–0.74], *p* = 0.002), NLR of <5 vs. ≥5 (HR: 0.65 [95% CI: 0.43–0.98], *p* = 0.04), PNI of ≥40 vs. <40 (HR: 0.38 [95% CI: 0.25–0.57], *p* < 0.001), and the number of induction cycles (4–6 vs. 1–3 cycles; HR: 0.30 [95% CI: 0.19–0.49], *p* < 0.001). Meanwhile, the dose reduction at initiating the first chemoimmunotherapy cycle did not reveal a significant correlation with PFS (HR: 1.04 [95% CI: 0.71–1.51], *p* = 0.84) or OS (HR: 1.21 [95% CI: 0.79–1.85], *p* = 0.39) as shown in Table 3.

As shown in Table 4, PFS outcomes among patients aged ≥75 were significantly associated with the number of induction chemoimmunotherapy cycles (4–6 vs. 1–3 cycles; HR: 0.35 [95% CI: 0.18–0.69], *p* = 0.003), and OS outcomes were significantly associated with ECOG-PS of 0–1 vs. ≥2 (HR: 0.38 [95% CI: 0.16–0.90], *p* = 0.03), NLR of <5 vs. ≥5 (HR: 0.40 [95% CI: 0.20–0.82], *p* = 0.01), PNI of ≥40 vs. <40 (HR: 0.28 [95% CI: 0.14–0.55], *p* < 0.001), and the number of induction cycles (4–6 vs. 1–3 cycles; HR: 0.25 [95% CI: 0.12–0.52], *p* < 0.001). On the other hand, OS outcomes among non-elderly patients (Table 5) were significantly associated with ECOG-PS of 0–1 vs. ≥2 (HR: 0.48 [95% CI: 0.24–0.96], *p* = 0.03), the number of induction chemoimmunotherapy cycles (4–6 vs. 1–3 cycles; HR: 0.37 [95% CI: 0.20–0.70], *p* = 0.002), and PNI of ≥40 vs. <40 (HR: 0.45 [95% CI: 0.27–0.76], *p* = 0.002), whereas PFS outcomes were significantly associated the number of induction chemoimmunotherapy cycles (4–6 vs. 1–3 cycles; HR: 0.41 [95% CI: 0.23–0.75], *p* = 0.003).

Multivariate analyses demonstrated that age (<75 vs. ≥75), sex, smoking status, baseline ECOG-PS, and dose reduction at initiating the first chemoimmunotherapy cycle did not correlate with PFS or OS. In contrast, the number of induction cycles (4–6 vs. 1–3 cycles; HR: 0.45 [95% CI: 0.27–0.73], *p* = 0.001) was associated with PFS. Moreover, baseline PNI of ≥40 vs. <40 (HR: 0.36 [95% CI: 0.22–0.59], *p* < 0.001) and the number of induction cycles (4–6 vs. 1–3 cycles; HR: 0.31 [95% CI: 0.18–0.54], *p* < 0.001) were related to OS (Table 3).

### 3.7. Relationship between Eastern Cooperative Oncology Group Performance Status at Disease Progression or at Second-Line Therapy Initiation and the Post-Progression Survival after First-Line Chemoimmunotherapy (Figure 5)

Among 155 patients, 112 (72.3%) experienced disease progression, and ECOG-PS at disease progression was available in 96 patients. The number of patients with ECOG-PS of 0, 1, 2, and 3–4 was 15, 52, 9, and 20, respectively. The median PPS of those who experienced disease progression with ECOG-PS of 0–1, 2, and 3–4 was 10.7 (95% CI: 9.0–13.8), 2.1 (95% CI: 0.4–8.6), and 2.1 months (95% CI: 1.1–2.8), respectively (*p* < 0.001).

Among the 112 patients, 99 (88.4%) underwent subsequent second-line therapy. The ECOG-PS at the second-line therapy initiation was available in 79 patients; the number of patients with ECOG-PS of 0, 1, 2, and 3–4 was 15, 52, 9, and 3, respectively. The PPS of patients who underwent second-line therapy with ECOG-PS of 0, 1, 2, and 3–4 was 16.4 (95% CI: 10.1-not applicable), 9.0 (95% CI: 7.9–12.5), 2.1 (95% CI: 0.4–8.6), and 2.8 months (95% CI: 1.6-not applicable), respectively (*p* < 0.001). There was a substantial difference in the PPS between those with an ECOG-PS of 0 and 1 at the second-line therapy initiation.

In total, 15 patients with an ECOG-PS of 0 at the second-line therapy initiation included 4 elderly patients. Dose reduction at first-line chemoimmunotherapy initiation was performed in 3 elderly patients (75.0%), whereas all 11 non-elderly patients were treated without dose reduction. On the other hand, 52 patients with an ECOG-PS of 1 included 18 elderly patients, in which preliminary dose reduction at the first-line therapy initiation was conducted in only eight patients (44.4%). The preliminary dose reduction at the first-line therapy initiation in elderly patients might have led to a reduction in AEs and the preservation of ECOG-PS at second-line therapy initiation, resulting in prolonged PPS.

## 4. Discussion

Age is not considered a prognostic factor in several randomized NSCLC studies [32,33,34,35]. However, early death within 6 months after first-line chemotherapy in elderly patients with cancer aged > 70 years is associated with malnutrition and poor mobility [36]. Furthermore, a functional decline in daily activities occurs in 16.7% [37] and 19.9% [38] of the patients after first-line chemotherapy. Similar results have been reported for lung cancer, with 23% of elderly patients aged ≥ 70 years experiencing a functional decline in daily activities after first-line chemotherapy [39]. Therefore, phase III trial results cannot be routinely applied to the daily clinic of elderly patients with cancer without modification.

A geriatric assessment (GA) is recommended to identify the vulnerabilities of patients aged ≥65 to overcome this age-related obstacle. CARG or Chemotherapy Risk Assessment Scale for High-Age Patients (CRASH) scoring tools are also recommended to predict chemotherapy toxicity risk [40,41]. A cluster-randomized study revealed that GA intervention reduced grade 3–5 toxicity without shortening the 6-month OS [42]. However, no tool is available for the prediction of the irAE risk in elderly patients, and the validity of chemoimmunotherapy in elderly patients with ES-SCLC remains unclear.

The introduction of ICIs has revolutionized the treatment strategy for ES-SCLC. Anti-PD-L1 antibodies of atezolizumab (IMpower133 trial) or durvalumab (CASPIAN trial) with platinum plus ETP regimens have improved PFS and OS compared to platinum plus ETP regimens [6,7]. In contrast, adding anti-programmed cell death 1 (PD-1) antibody to pembrolizumab in the KEYNOTE-604 trial improved PFS more compared to the placebo. However, it did not improve OS [43]. Although irAE risk is unpredictable, chemoimmunotherapy’s favourable effects on cancer-related symptoms and quality of life (QOL) in patients with ES-SCLC have been demonstrated in IMpower133 [44] and CASPIAN trials [45]. Thus, it is crucial to understand whether chemoimmunotherapy could improve PFS, OS, cancer-related symptoms, and QOL in elderly patients with ES-SCLC.

This multicenter retrospective study elucidated that the same efficacy of chemoimmunotherapy could be expected in elderly and non-elderly patients regarding ORR, PFS, OS, and PPS. Furthermore, AEs and irAEs did not exhibit a higher increase in elderly patients compared with those in non-elderly patients. Thus, chemoimmunotherapy for ES-SCLC would benefit elderly patients aged ≥ 75. This information can facilitate the administration of chemoimmunotherapy in elderly patients with ES-SCLC, especially in those who avoided chemoimmunotherapy due to age and patient preference.

Whereas dose reduction at the first cycle initiation was more frequent in elderly patients (*p* = 0.03), the efficacy of chemoimmunotherapy was similar in non-elderly patients, suggesting that the physician’s cautious patient-specific dose reduction would not negatively affect these outcomes. Four or more chemoimmunotherapy induction cycles were associated with prolonged PFS (HR: 0.45) and OS (HR: 0.31). This finding suggests the importance of platinum plus ETP treatment in managing ES-SCLC. Moreover, substantial tumour reduction before continuing maintenance therapy with an anti-PD-L1 antibody could improve PFS and OS, considering the aggressive nature of SCLC. Thus, the results revealed that the standard four-cycle induction chemoimmunotherapy would be optimal for treating ES-SCLC.

The significantly high correlation between PPS and OS compared to the correlation between PFS and OS was also demonstrated in this study. The PPS of patients who experienced disease progression with an ECOG-PS of 0–1 was significantly longer than that of patients with ECOG-PS ≥ 2 (*p* < 0.001). Furthermore, the PPS of patients who underwent second-line therapy was significantly longer with an ECOG-PS of 0 at second-line therapy initiation than in those with an ECOG-PS of 1 (*p* < 0.001).

These results suggest that reducing AEs and maintaining the ECOG-PS during first-line chemoimmunotherapy by appropriate individual management, including dose reduction and supportive care, is crucial for PPS improvement in patients who proceed to second-line therapy.

Immunological and nutritional markers are considered to reflect the close relationship between cancer treatment and local or systemic host–tumour interactions. Thus, these markers were also investigated as candidate biomarkers for predicting treatment outcomes. PLR [16], NLR [17,18], PNI [19,20], and SII [21] are predictive of OS in patients with ES-SCLC. NLR and PNI significantly correlated with OS in this study. However, univariate analyses among elderly and non-elderly patients did not show any age-specific markers.

The higher rate of receiving second-line therapy after disease progression (88.4%) was a significant feature in this study, compared to the previously reported real-world evidence in Canada where only 8.7% of the patients underwent second-line therapy [46]. The data in Canada were collected before the introduction of chemoimmunotherapy for ES-SCLC, and a real-world evaluation of atezolizumab in combination with platinum-etoposide chemotherapy in Canada reported that 27% of patients treated with chemoimmunotherapy received second-line therapy, whereas 15% of patients treated with chemotherapy proceeded to second-line therapy [47]. Thus, the treatment strategy could be evolving after the introduction of chemoimmunotherapy. In addition, the observed feature could be partly explained by the national traits.

This study had several limitations. First, its retrospective design was prone to bias. Second, despite the relatively large sample size, it was insufficient for category-stratified analysis, especially in the ECOG-PS status at disease progression or at second-line therapy initiation. Univariate analyses were used to assess the age-stratified analysis of PFS and OS outcomes, which is another potential source of bias. Third, we selected cut-off values for various predictive biomarkers from previous reports. Therefore, using biomarkers and optimal cut-off values should be clarified in large prospective studies. Finally, this study evaluated the age-stratified PFS and OS outcomes among the patients treated with chemoimmunotherapy, which resulted in an indirect comparison between elderly and non-elderly patients. Because it is difficult to retrospectively compare the chemoimmunotherapy and chemotherapy groups, the indirect comparison was used in this study.

## 5. Conclusions

This study revealed that chemoimmunotherapy would benefit elderly patients with ES-SCLC aged ≥ 75 years, with PFS and OS improvement similar to that in non-elderly patients. The observed AEs and irAEs were tolerable in elderly patients, facilitating chemoimmunotherapy application in elderly patients. An ECOG-PS of 0 at subsequent second-line therapy initiation predicted a substantially longer PPS than an ECOG-PS of 1. This suggests that ECOG-PS maintenance by appropriate individual management during first-line chemoimmunotherapy is crucial in the management of ES-SCLC. Thus, elderly patients with ES-SCLC should be cautiously treated to improve outcomes.

## Figures and Tables

**Figure 1 cancers-15-01543-f001:**
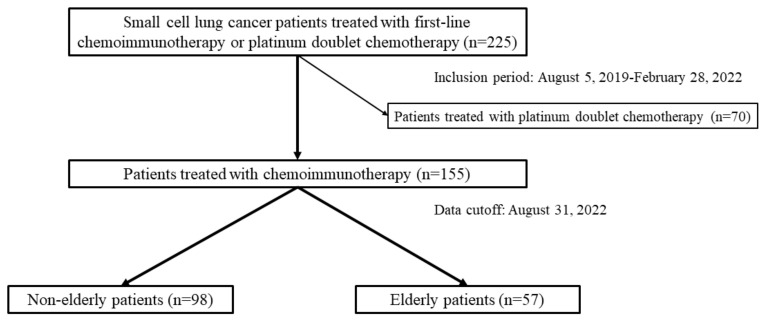
Diagram presenting patient selection among 225 consecutive patients with small cell lung cancer who underwent first-line therapy between 5 August 2019 and 28 February 2022; 155 of them received chemoimmunotherapy. Age-stratified analyses were conducted to compare non-elderly patients aged <75 (*n* = 98) and elderly patients aged ≥75 (*n* = 57).

**Figure 2 cancers-15-01543-f002:**
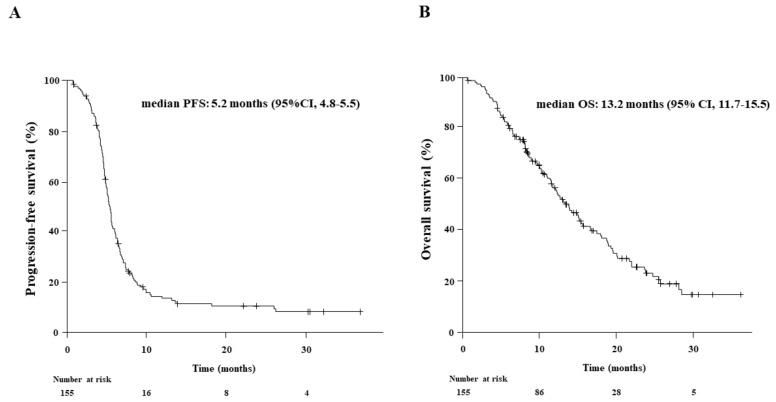
Kaplan-Meier estimates of (**A**) progression-free survival (PFS) and (**B**) overall survival (OS) in 155 patients who received first-line chemoimmunotherapy. The median PFS and OS were 5.2 and 13.2 months, respectively.

**Figure 3 cancers-15-01543-f003:**
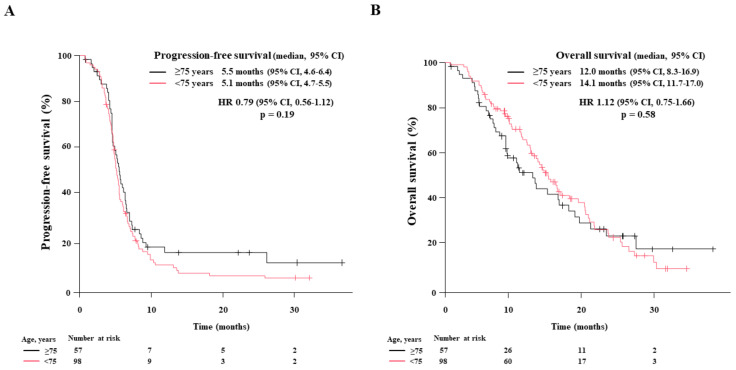
Age-stratified survival of patients treated with first-line chemoimmunotherapy. The Kaplan-Meier estimates of (**A**): progression-free survival (PFS) and (**B**): overall survival (OS) in non-elderly patients < 75 years (*n* = 98) and elderly patients aged ≥75 (*n* = 57) are illustrated. The median PFS and OS are presented with a 95% confidence interval (95% CI).

**Figure 4 cancers-15-01543-f004:**
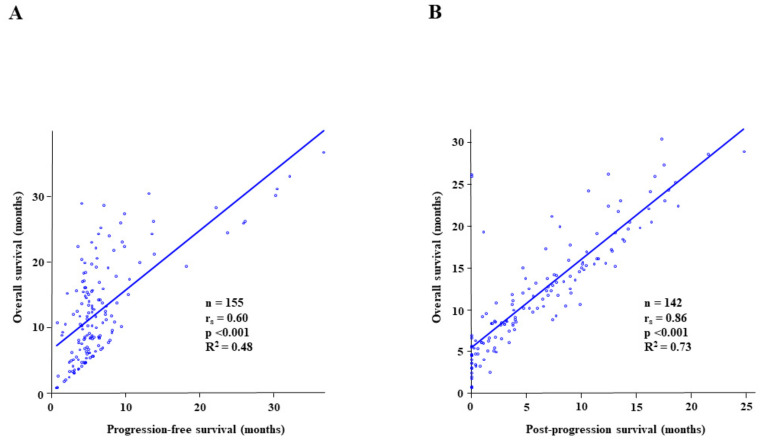
The Spearman’s rank correlation analysis and linear regression analysis to reveal (**A**) a correlation between progression-free survival (PFS) and overall survival (OS); (**B**) a correlation between post-progression survival (PPS) and OS. The r_s_ value and the R^2^ value represent Spearman’s rank correlation coefficient and linear regression, respectively.

**Figure 5 cancers-15-01543-f005:**
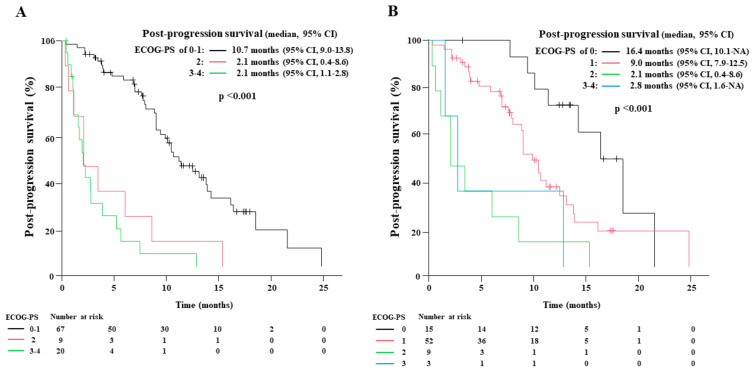
The Kaplan-Meier estimates of post-progression survival (PPS) according to the Eastern Cooperative Oncology Group performance status (ECOG-PS) at the disease progression of first-line chemoimmunotherapy (**A**), and at the start of second-line therapy (**B**).

**Table 1 cancers-15-01543-t001:** Characteristics of patients treated with platinum doublet chemotherapy.

Characteristics	Total: *n* = 70	Non-Elderly Patients (<75 Years): *n* = 26	Elderly Patients (≥75 Years): *n* = 44	*p*-Value
Sex: male/female	53 (75.7%)/17 (24.3%)	16 (61.5%)/10 (38.5%)	37 (84.1%)/7 (15.9%)	0.05
Age (years): median (range)	76 (61–86)	70 (61–73)	78 (75–86)	N/A
ECOG-PS: 0/1/2/3/4	7 (10.0%)/40 (57.1%)/11 (15.7%)/12 (17.1%)/0	3 (11.5%)/13 (50.0%)/2 (7.7%)/8 (30.8%)/0	4 (9.1%)/27 (61.4%)/9 (20.5%)/4 (9.1%)/0	0.09
Regimen				
CBDCA + etoposide	64 (91.4%)	20 (76.9%)	44 (100%)	** 0.002 **
CDDP + etoposide	3 (4.3%)	3 (11.5%)	0	** 0.04 **
CDDP + irinotecan	3 (4.3%)	3 (11.5%)	0	** 0.04 **
Reasons for adopting chemotherapy				
Patient’s preference	5 (7.1%)	5 (19.2%)	0	** 0.005 **
Interstitial lung disease	33 (47.1%)	11 (42.3%)	22 (50.0%)	0.62
Poor ECOG-PS	13 (18.6%)	6 (23.1%)	7 (15.9%)	0.53
Age	8 (11.4%)	1 (3.8%)	7 (15.9%)	0.24
Rheumatoid arthritis	2 (2.9%)	2 (7.7%)	0	0.14
IgA nephritis	1 (1.4%)	1 (3.8%)	0	0.37
Dermatomyositis	1 (1.4%)	0	1 (2.3%)	1.00
Congestive heart failure	3 (4.3%)	0	3 (6.8%)	0.29
Sick sinus syndrome	1 (1.4%)	0	1 (2.3%)	1.00
Cognitive impairment	1 (1.4%)	0	1 (2.3%)	1.00
Liver dysfunction	1 (1.4%)	0	1 (2.3%)	1.00
Renal dysfunction	1 (1.4%)	0	1 (2.3%)	1.00

The bold and underline in the tables means the statistical significance.

**Table 2 cancers-15-01543-t002:** Characteristics of patients treated with chemoimmunotherapy.

Characteristics	Total: *n* = 155	Non-Elderly Patients (<75 Years): *n* = 98	Elderly Patients (≥75 Years): *n* = 57	*p*-Value
Sex: male/female	125 (80.6%)/30 (19.4%)	74 (75.5%)/24 (24.5%)	51 (89.5%)/6 (10.5%)	** 0.04 **
Age (years): median (range)	72 (43–87)	69 (43–74)	79 (75–87)	
ECOG-PS: 0/1/2/3/4	39 (25.2%)/94 (60.6%)/17 (11.0%)/5 (3.2%)/0	32 (32.7%)/52 (53.1%)/11 (11.2%)/3 (3.1%)/0	7 (12.3%)/42 (73.7%)/6 (10.5%)/2 (3.5%)/0	** 0.03 **
Smoking: current/former/never	54 (34.8%)/96 (61.9%)/5 (3.2%)	38 (38.8%)/55 (56.1%)/5 (5.1%)	16 (28.1%)/41 (71.9%)/0	** 0.04 **
Stage: III/IVA/IVB/recurrence	9 (5.8%)/40 (25.8%)/103 (66.5%)/3 (1.9%)	5 (5.1%)/22 (22.4%)/68 (69.4%)/3 (3.1%)	4 (7.0%)/18 (31.6%)/35 (61.4%)/0	0.35
Brain metastasis at baseline: (+)/(—)	33 (21.3%)/122 (78.7%)	25 (25.5%)/73 (74.5%)	8 (14.0%)/49 (86.0%)	0.11
Liver metastasis at baseline: (+)/(—)	37 (23.9%)/118 (76.1%)	24 (24.5%)/74 (75.5%)	13 (22.8%)/44 (77.2%)	0.85
ILD at baseline: (+)/(—)	6 (3.9%)/149 (96.1%)	2 (2.0%)/96 (98.0%)	4 (7.0%)/53 (93.0%)	0.19
Autoimmune disease: (+)/(—)	2 (1.3%)/153 (98.7%)	1 (1.0%)/97 (99.0%)	1 (1.8%)/56 (98.2%)	1.00
Steroid treatment at baseline: (+)/(—)	7 (4.5%)/148 (95.5%)	7 (7.1%)/91 (92.9%)	0/57 (100%)	** 0.04 **
Cycles of induction: 1–2/3/4/5–6	13 (8.4%)/13 (8.4%)/126 (81.3%)/3 (1.9%)	4 (4.1%)/9 (9.2%)/83 (84.7%)/2 (2.0%)	9 (15.8%)/4 (7.0%)/43 (75.4%)/1 (1.8%)	0.08
Reduced dose at 1st cycle: (+)/(—)	45 (29.0%)/110 (71.0%)	20 (20.4%)/78 (79.6%)	25 (43.9%)/32 (56.1%)	** 0.003 **
Dose reduction during Tx.: (+)/(—)	62 (40.0%)/93 (60.0%)	35 (35.7%)/63 (64.3%)	27 (47.4%)/30 (52.6%)	0.18
G-CSF during induction Tx.: (+)/(—)	91 (58.7%)/64 (41.3%)	58 (59.2%)/40 (40.8%)	33 (57.9%)/24 (42.1%)	1.00
Reason for discontinuation of 1st-line				
Disease progression	112 (72.3%)	80 (81.6%)	32 (56.1%)	** <0.001 **
Adverse events	3 (1.9%)	1 (1.0%)	2 (3.5%)	0.55
Immune-related adverse events	9 (5.8%)	5 (5.1%)	4 (7.0%)	0.73
Others	18 (11.6%)	5 (5.1%)	13 (22.8%)	** 0.001 **
Ongoing Tx.	13 (8.4%)	7 (7.1%)	6 (10.5%)	0.55
2nd-line Tx.: (+)/(—)/on 1st-line Tx.	99 (63.9%)/43 (27.7%)/13 (8.4%)	70 (71.4%)/21 (21.4%)/7 (7.1%)	29 (50.9%)/22 (38.6%)/6 (10.5%)	** 0.04 **

The bold and underline in the tables means the statistical significance.

**Table 3 cancers-15-01543-t003:** Univariate and multivariate analyses to predict progression-free survival and overall survival among 155 patients treated with chemoimmunotherapy.

	Progression-Free Survival						Overall Survival					
	Univariate Analysis			Multivariate Analysis			Univariate Analysis			Multivariate Analysis		
Variables	HR	95% CI	*p*-Value	HR	95% CI	*p*-Value	HR	95% CI	*p*-Value	HR	95% CI	*p*-Value
Age (<75/≥75)	0.79	0.56–1.12	0.19	0.81	0.55–1.19	0.28	1.12	0.75–1.66	0.58	1.18	0.76–1.84	0.47
Sex (female/male)	1.25	0.82–1.90	0.31	1.18	0.73–1.91	0.49	1.23	0.73–2.06	0.45	1.05	0.60–1.85	0.86
Smoking (—/+)	1.21	0.49–2.95	0.68	1.53	0.50–4.72	0.45	1.68	0.61–4.59	0.31	1.92	0.58–6.34	0.29
ECOG-PS (<2/≥2)	0.94	0.58–1.54	0.82	0.75	0.42–1.32	0.32	** 0.43 **	** 0.25–0.74 **	** 0.002 **	0.55	0.30–1.03	0.06
Reduced dose at 1st cycle (—/+)	1.04	0.71–1.51	0.84	0.94	0.63–1.39	0.75	1.21	0.79–1.85	0.39	0.97	0.62–1.54	0.90
Cycles of induction (≥4/<4)	** 0.38 **	** 0.25–0.59 **	** <0.001 **	** 0.45 **	** 0.27–0.73 **	** 0.001 **	** 0.30 **	** 0.19–0.49 **	** <0.001 **	** 0.31 **	** 0.18–0.54 **	** <0.001 **
PLR (<250/≥250)	0.92	0.65–1.30	0.62				0.88	0.59–1.32	0.55			
NLR (<5/≥5)	0.72	0.50–1.02	0.07	0.84	0.55–1.30	0.44	** 0.65 **	** 0.43–0.98 **	** 0.04 **	0.96	0.58–1.58	0.84
PNI (≥40/<40)	** 0.64 **	** 0.45–0.92 **	** 0.02 **	0.70	0.46–1.07	0.10	** 0.38 **	** 0.25–0.57 **	** <0.001 **	** 0.36 **	** 0.22–0.59 **	** <0.001 **
SII (<1000/≥1000)	0.77	0.55–1.08	0.13				0.89	0.61–1.30	0.55			

The bold and underline in the tables means the statistical significance.

**Table 4 cancers-15-01543-t004:** Univariate analysis to predict progression-free survival and overall survival among elderly patients aged ≥ 75.

	Univariate Analysis (≥75 Years)					
	Progression-Free Survival			Overall Survival		
Variables	HR	95% CI	*p*-Value	HR	95% CI	*p*-Value
Sex (female/male)	1.24	0.49–3.14	0.65	1.68	0.51–5.47	0.39
ECOG-PS (<2/≥2)	0.79	0.35–1.78	0.57	** 0.38 **	** 0.16–0.90 **	** 0.03 **
Reduced dose at 1st cycle (—/+)	1.06	0.60–1.87	0.85	1.07	0.56–2.05	0.84
Cycles of induction (≥4/<4)	** 0.35 **	** 0.18–0.69 **	** 0.003 **	** 0.25 **	** 0.12–0.52 **	** <0.001 **
PLR (<250/≥250)	0.92	0.51–1.68	0.79	0.87	0.44–1.71	0.68
NLR (<5/≥5)	0.59	0.32–1.09	0.09	** 0.40 **	** 0.20–0.82 **	** 0.01 **
PNI (≥40/<40)	0.58	0.32–1.05	0.07	** 0.28 **	** 0.14–0.55 **	** <0.001 **
SII (<1000/≥1000)	0.93	0.52–1.65	0.79	0.84	0.44–1.59	0.59

The bold and underline in the tables means the statistical significance.

**Table 5 cancers-15-01543-t005:** Univariate analysis to predict progression-free survival and overall survival among non-elderly patients < 75 years.

	Univariate Analysis (<75 Years)					
	Progression-Free Survival			Overall Survival		
Variables	HR	95% CI	*p*-Value	HR	95% CI	*p*-Value
Sex (female/male)	1.29	0.80–2.09	0.30	1.05	0.58–1.90	0.88
ECOG-PS (<2/≥2)	1.28	0.70–2.35	0.43	** 0.48 **	** 0.24–0.96 **	** 0.03 **
Reduced dose at 1st cycle (-/+)	0.99	0.57–1.71	0.98	1.26	0.68–2.33	0.46
Cycles of induction (≥4/<4)	** 0.41 **	** 0.23–0.75 **	** 0.003 **	** 0.37 **	** 0.20–0.70 **	** 0.002 **
PLR (<250/≥250)	1.18	0.76–1.81	0.46	0.91	0.55–1.49	0.70
NLR (<5/≥5)	0.80	0.51–1.24	0.32	0.83	0.50–1.39	0.48
PNI (≥40/<40)	0.77	0.49–1.19	0.24	** 0.45 **	** 0.27–0.76 **	** 0.002 **
SII (<1000/≥1000)	1.48	0.97–2.27	0.07	1.36	0.84–2.21	0.21

The bold and underline in the tables means the statistical significance.

## Data Availability

The datasets generated during the current study are available from the corresponding author on reasonable request.

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
