# Peer review of "Age-Stratified Analysis of First-Line Chemoimmunotherapy for Extensive-Stage Small Cell Lung Cancer: Real-World Evidence from a Multicenter Retrospective Study"

_cancers, 2023, doi:10.3390/cancers15051543_

Round 1

Reviewer 1 Report

In this study, Takeda et al., evaluated survival outcomes of extensive-stage small cell lung cancer patients in Japan who received first-line chemoimmunotherapy stratified by age (<75 vs. ³ 75 years). The authors observed that the elderly age group (³ 75)  had similar progression-free and overall survival as the non-elderly group (< 75). The results of this study are important and warrant publication. The manuscript could benefit from some minor changes and potentially some additional analyses (please see below). 

Abstract

-       I would be cautious in saying that ECOG-PS was not correlated with PFS or OS. It wasn’t statistically significantly associated, but the effect size was actually pretty strong and the confidence interval barely crossed the null, which could be a function of a small sample size. In addition, ECOG was compared between <2 and ³ 2. If 0 was compared to ³ 2 it might actually be significant. 

Introduction

-       Line 72 – it doesn’t make sense to start that sentence with “Additionally”. 

-       Line 77 - What is the geographic location for the statistic from reference 8? This is important to contextualize. 

-       Please include a reference for post-progression survival. 

Methods

-       Throughout the paper the authors say “univariate” or “multivariate” when they are actually referring to “univariable” and “multivariable” analyses. Univariate or multivariate refer to multiple outcomes, but only one outcome is being examined at a time. Please correct this throughout the manuscript. 

Results

-       Line 223 – reasons why chemoimmunotherapy was considered unfit are not presented in Table 1, so it should not be in brackets at the end of this sentence. 

-       It would be useful to compare characteristics of patients that received chemoimmunotherapy versus just chemotherapy. Please consider including this analysis.

-       It would also be useful to list difference in characteristics between elderly and non-elderly in the text, rather than just providing it in the table. 

-       Line 303 – this is an interpretation of results. Please remove and include in the discussion.

-       Line 311 – Age-specific marker? What does that mean?

-       I would suggest conducting multivariable analyses of prognostic factors for both <75 and ³ 75. It appears that there could be a difference for NLR and PNI – I would suggest testing if there is an interaction by age. If those are unique prognostic factors for ³ 75 that would be an important finding. 

Discussion

-       An important finding of this study is the high rate of uptake of second-line therapy among patients that received first-line chemoimmunotherapy. In the context of other real-world studies that have observed very low uptake of second-line therapy among patients that received first-line chemotherapy (PMID: 34436036), this is an important finding. Some acknowledgement of this in the discussion is warranted. 

-       Last paragraph - Please elaborate on the potential biases. 

Author Response

Response to the Reviewer 1:

I really appreciate your sincere suggestions and advice based on the deep understanding of our manuscript. I have read through all of your comments and revised the manuscript in accordance with your suggestions.

I have responded to your comments in a point-by-point manner below.

Abstract

-       I would be cautious in saying that ECOG-PS was not correlated with PFS or OS. It wasn’t statistically significantly associated, but the effect size was actually pretty strong and the confidence interval barely crossed the null, which could be a function of a small sample size. In addition, ECOG was compared between <2 and ³ 2. If 0 was compared to ³ 2 it might actually be significant. 

Response:

I appreciate your sincere suggestion, and I understand your concern, because ECOG-PS is one of the strongest predictive markers of OS. Therefore, I avoided referring to the relationship between ECOG-PS and PFS or OS in the Abstract section.

Introduction

-       Line 72 – it doesn’t make sense to start that sentence with “Additionally”. 

Response:

I appreciate your suggestion, and I switched “Additionally” to “and” in this part.

-       Line 77 - What is the geographic location for the statistic from reference 8? This is important to contextualize. 

Response:

I appreciate your suggestion, and I thought the relevant part should be more readable.

Thus, I changed the reference No.8 from the European data to the US data, presenting the median age of those patients who were diagnosed with lung cancer in the US.

-       Please include a reference for post-progression survival. 

Response:

The reference for post-progression survival is reference No.23, in the last part of Introduction section.

Methods

-       Throughout the paper the authors say “univariate” or “multivariate” when they are actually referring to “univariable” and “multivariable” analyses. Univariate or multivariate refer to multiple outcomes, but only one outcome is being examined at a time. Please correct this throughout the manuscript. 

Response:

I really appreciate your comments, and discussed with my colleagues who dealt with the statistical analysis. They used the “univariate” and “multivariate” analyses in EZR (reference No. 31), and we concluded that these expressions would be better maintaining the status quo.

Results

-       Line 223 – reasons why chemoimmunotherapy was considered unfit are not presented in Table 1, so it should not be in brackets at the end of this sentence. 

Response:

I thank you for your comments. The reasons why chemoimmunotherapy were considered unfit are presented in Table 1 as “Reasons for adopting chemotherapy”.

-       It would be useful to compare characteristics of patients that received chemoimmunotherapy versus just chemotherapy. Please consider including this analysis.

Response:

I appreciate your suggestions, and agree with your opinion.

However, we omitted the detailed baseline characteristics of patients who received chemotherapy during the conceptualization of this study, in order to minimize the efforts needed to fill in the case report forms. Thus, we simplified the baseline characteristics to sex, age, ECOG-PS, regimens, reasons for adopting chemotherapy, which are presented in Table 1.

-       It would also be useful to list difference in characteristics between elderly and non-elderly in the text, rather than just providing it in the table. 

Response:

I appreciate your suggestions and added the description in the text (Lines 221-223).

-       Line 303 – this is an interpretation of results. Please remove and include in the discussion.

Response:

I appreciate your suggestion, and I removed the relevant expression and included in the Discussion section (Lines 401-403).

-       Line 311 – Age-specific marker? What does that mean?

Response:

I thank you for the suggestion, and I deleted the relevant sentence.

-       I would suggest conducting multivariable analyses of prognostic factors for both <75 and ³ 75. It appears that there could be a difference for NLR and PNI – I would suggest testing if there is an interaction by age. If those are unique prognostic factors for ³ 75 that would be an important finding. 

Response:

I really appreciate your suggestions, and I agree with you in that the addition of analysis for patients aged <75. However, the multivariate analyses for patients aged <75 or ≥75 were impossible because of the small sample size. Thus, we added a univariate analysis to predict PFS and OS among non-elderly patients aged <75 and presented the data in Table 3.C. The relevant description was also added in the Results section (Lines 316-322) and Discussion section (Line 427).

Discussion

-       An important finding of this study is the high rate of uptake of second-line therapy among patients that received first-line chemoimmunotherapy. In the context of other real-world studies that have observed very low uptake of second-line therapy among patients that received first-line chemotherapy (PMID: 34436036), this is an important finding. Some acknowledgement of this in the discussion is warranted. 

Response:

I really appreciate your sincere advice and suggestion. I recognize this point important, and added sentences in the Discussion section (Lines 429-438).

-       Last paragraph - Please elaborate on the potential biases. 

Response:

I thank you for the sincere suggestions. I added some sentences in the Discussion section (Lines 442-443, 445-449).

Reviewer 2 Report

Chemoimmunotherapy treatment efficacy and safety in elderly patients ≥75 years with ES-SCLC was analysed  through real-world Japanese evidence. The article is well presented and the results are significant. 

Minor comments:

I would improve the abstract background. It is too direct, please write a bit more of background on the subject. Please, add titles beside the figure numbers and then keep the legends. The figures are showing legends directly, without figure titles.

Tables should also present the descriptive titles.

Author Response

I really appreciate your deep understanding of our manuscript as well as sincere suggestions and advice.

I revised the abstract section by deleting the definitive adjectives. And I also added the titles in the relevant figures and tables.
